# Availability of Metribuzin-Loaded Polymeric Nanoparticles in Different Soil Systems: An Important Study on the Development of Safe Nanoherbicides

**DOI:** 10.3390/plants11233366

**Published:** 2022-12-04

**Authors:** Vanessa Takeshita, Gustavo Vinicios Munhoz-Garcia, Camila Werk Pinácio, Brian Cintra Cardoso, Daniel Nalin, Valdemar Luiz Tornisielo, Leonardo Fernandes Fraceto

**Affiliations:** 1Center of Nuclear Energy in Agriculture, University of São Paulo, Av. Centenário 303, Piracicaba 13400-970, SP, Brazil; 2Institute of Science and Technology, São Paulo State University (UNESP), Av. Três de Março 511, Sorocaba 18087-180, SP, Brazil

**Keywords:** nanoformulation, nanoherbicide, sorption–desorption, soil mobility, soil organic matter

## Abstract

Nanoformulations have been used to improve the delivery of fertilizers, pesticides, and growth regulators, with a focus on more sustainable agriculture. Nanoherbicide research has focused on efficiency gains through targeted delivery and environmental risk reduction. However, research on the behavior and safety of the application of these formulations in cropping systems is still limited. Organic matter contained in cropping systems can change the dynamics of herbicide–soil interactions in the presence of nanoformulations. The aim of this study was to use classical protocols from regulatory studies to understand the retention and mobility dynamics of a metribuzin nanoformulation, compared to a conventional formulation. We used different soil systems and soil with added fresh organic material. The batch method was used for sorption–desorption studies and soil thin layer chromatography for mobility studies, both by radiometric techniques. Sorption parameters for both formulations showed that retention is a reversible process in all soil systems (H~1.0). In deep soil with added fresh organic material, nanoformulation was more sorbed (14.61 ± 1.41%) than commercial formulation (9.72 ± 1.81%) (*p* < 0.05). However, even with the presence of straw as a physical barrier, metribuzin in nano and conventional formulations was mobile in the soil, indicating that the straw can act as a barrier to reduce herbicide mobility but is not impeditive to herbicide availability in the soil. Our results suggest that environmental safety depends on organic material maintenance in the soil system. The availability can be essential for weed control, associated with nanoformulation efficiency, in relation to the conventional formulation.

## 1. Introduction

Formulations that improve the agricultural performance of pesticides and cause less environmental impact are goals of modern agriculture. Traditional formulations are applied in high amounts per area, presenting the potential for the selection of resistant organisms and efficiency loss; furthermore, a large part can be lost to the environment [1]. Nanotechnology has been explored in several fields of knowledge, such as medicine, electronics, cosmetics, food, and agriculture [2]. In agriculture, formulations can be applied in different ways, to improve the efficiency of fertilizers, pesticides, and growth regulators, to promote sustainable agriculture, with the potential for and necessity to be explored even further [3]. Nanoformulations have the potential to minimize the impacts of pesticides on ecosystems, protect molecules, and increase efficiency in pest and disease control [4,5,6,7,8]. Studies with pesticide nanoformulations showed minimization of active losses due to degradation, volatilization, and leaching [7,9,10,11], as well as alterations in soil retention and mobility [12,13,14], and increases in efficiency and action in plants [6,15].

Among pesticides, the association of herbicides with nanoformulations has been widely explored in recent years [6,13,15,16]. One of the motivations for nanoherbicide research is the worldwide application on a large scale (40% of the 40 million tons of pesticides per year) in relation to other pesticides [17]. The use of nanoformulations of pre-emergent herbicides, as a recovery strategy against glyphosate-tolerant and resistant plants in the field [18,19], is an emerging trend. In addition, pesticides can be largely lost to the environment, with only 1% remaining in the target organism [1]; this is a concern for herbicides applied directly to the soil, such as pre-emergent herbicides.

Nanoformulation of metribuzin (*nano*MTZ), a pre-emergent triazine herbicide, has been one of the focuses of study in our research group. In a conventional formulation, metribuzin presents high solubility in water (1.05 g L^−1^ at 20 °C) and is weakly adsorbed by soil particles, therefore, presenting great potential for deep mobility in the soil profile [20]. Metribuzin is frequently found in groundwater around the world, which increases the risks to human health and environmental quality [20,21,22]. Associated with PCL (poly-ε-caprolactone) nanoparticles, metribuzin (MTZ) showed increased effectiveness against weed plants, while it showed no effects on the enzymatic activity of different soils, and no increased persistence in the environment [23].

Metribuzin is a photosystem II inhibitor, used for broad and grass weed control in some crop systems, such as sugarcane, soybean, wheat, and corn [24,25]. These agricultural systems can be cultivated through conventional tillage and non-tillage. The no-tillage system adds organic residues from the previous crop to the soil and promotes an increase in soil organic matter (SOM) over time [26,27]. On the other hand, the conventional system, with greater exposure and degradation of organic residues, due to soil disturbance, promotes less accumulation of SOM [28]. In crop systems with high SOM content, sorption capacity is increased and pesticide mobility is reduced [29,30]. However, for nanoformulations, behavior in soil crop systems has not yet been explored. Understanding the effects of soil properties, especially organic matter, is essential for the adoption and safe use of nanoformulations in agriculture, such as the nanoformulation of metribuzin applied directly to the soil.

Sorption and desorption experiments were carried out in different crop system soils to compare the retention and availability of a conventional and a nano metribuzin formulation. Consolidated cropping systems of more than 15 years and soils with added fresh organic matter were chosen for the study. In the same way, soil mobility studies were used to verify the short distance traveled by formulations in the soil plates. Herein, we showed that *nano*MTZ can be used more safely than conventional MTZ when applied in soil with the addition of fresh organic matter and depending on the system’s complexity. The results of this study can contribute to the development of better nanoherbicides to be used in weed control, contributing to more sustainable agriculture.

## 2. Results

### 2.1. Metribuzin Nanoformulation and Commercial Formulation Sorption–Desorption in Different Soil Systems

The nanoMTZ and conventional MTZ sorption–desorption processes were influenced by the soil system (*p* < 0.05). A non-significant effect was observed for formulation (nano or conventional) and for the interaction between the factors (*p* > 0.05) (Appendix A). Non-cultivated (NC) soil was responsible for the highest ^14^C-metribuzin (63.27 ± 0.92%) sorption, followed by soybean–corn succession cultivated under no-till (SC-NT) (51.7 ± 2.23%). Soil cultivated with sugarcane monoculture (SG-MN) and soybean–corn succession cultivated under conventional tillage (SC-CT) presented less herbicide retention (49.13 ± 1.33% and 47.27 ± 1.2%, respectively) (Figure 1). Desorption was pronounced in all soil systems (39.38–41.9%), in relation to the control NC soil (29.87 ± 1.11%) (Figure 1). The sorption coefficient normalized to organic carbon (Koc) (Appendix A), presented higher values for SG-MN (120.41 ± 6.42 mL g^−1^) compared to other soil systems (SC-CT, SC-NT, and NC). In the desorption values, the Koc was greatest for SG-MN (198.98 ± 33.28 mL g^−1^), differently from NC (130.29 ± 14.46 mL g^−1^) (Appendix A).

The Freundlich sorption coefficient (Kf) indicates the intensity of ^14^C-metribuzin sorption in the soil of the different cropping systems and the value of 1/n indicates the linearity of the isotherm (Table 1 and Figure 2). The sorption and desorption isotherm slopes increase with increasing Kf values (Figure 2). The Kf sorption values ranged from 1.49–2.66 mL g^−1^ for *nano*MTZ and 1.33–2.73 mL g^−1^ for conventional MTZ (Table 1), while the Kf desorption values were between 1.39–3.17 mL g^−1^ and 0.26–3.78 mL g^−1^ for *nano*MTZ and conventional MTZ, respectively (Table 1). Freundlich sorption isotherm linearity (Figure 2) for *nano*MTZ and conventional MTZ showed values close to 1.0 (Table 1). These 1/n values (~1.0) represent type C curves [31], where sorption increases with increasing herbicide concentration, indicating that there is no saturation of soil sorption sites. Sorption and desorption isotherm linearity is used to calculate the hysteresis (H). Hysteresis indicates the reversibility of the sorption process and is related to the availability of the herbicide in the environment. Values of H > 0.7 indicate that the desorption process occurs at the same intensity as the sorption process [32], indicating that even after sorption, the herbicide easily returns to the soil solution, where it is available for transport, absorption, and degradation processes.

Pearson’s correlation matrices and correlation coefficients (r) between sorption–desorption of *nano*MTZ and conventional MTZ with soil properties are represented in Figure 3 (data can be consulted in Appendix A). For both formulations, sorption has a positive correlation with Kd sorption (r~0.99), Kd desorption (r~0.98), silt content (r~0.99), organic carbon (OC) (r = 0.96–0.98), and SOM (r = 0.96–0.98), and a negative correlation with desorption (r~−0.99). While desorption is negatively correlated with Kd desorption (r~−0.99), silt content (r~−0.98), OC (r = −0.98), and SOM (r = −0.98). Furthermore, for the nanoformulation, desorption is positively correlated with clay content (r = 0.98). For conventional MTZ, the clay content is negatively correlated with sorption (r = −0.96) and with Kd sorption (r = −0.96). No significant correlation between pH, cation exchange capacity (CEC), and the sum of bases (SB) was indicated by Pearson’s correlation matrices (Figure 3).

### 2.2. Organic Residue Effects on Sorption–Desorption of Metribuzin Nano and Conventional Formulations

The formulation and the organic residue type influenced metribuzin sorption and desorption in the soil independently (*p* < 0.05), with no interaction between the factors. The *nano*MTZ (14.61 ± 1.41%) presented 1.5-fold higher sorption than the MTZ (9.72 ± 1.81%) (Figure 4a, data can be consulted in Appendix A). *Nano*MTZ desorption was lower (70.52 ± 3.42%) than the conventional formulation (79.89 ± 2.74%) (Figure 4c). Metribuzin sorption in the soil without the addition of organic material was 10.88 ± 3.28% (Figure 4b). Organic materials were not very effective in increasing MTZ retention. The highest sorption occurred with the addition of sugarcane straw (13.37 ± 2.76%) and black oat straw (12.6 ± 2.63%). The forage turnip straw showed the lowest herbicide sorption (11.22 ± 3.28%). Desorption was higher for MTZ (79.89 ± 2.74%) compared to *nano*MTZ (70.53 ± 3.42%). In organic residues, desorption was higher in cassava (77.91 ± 2.4%) and corn (77.54 ± 5.76%), and the lowest desorption rates were observed in black oat (71.97 ± 4.32%) and sugarcane straw (73.18 ± 4.6%) (Figure 4d).

### 2.3. NanoMTZ and Conventional MTZ Soil Mobility

The retention factor (Rf) for *nano*MTZ and MTZ is shown in Appendix A. The Rf parameter aids the understanding of herbicide mobility in the soil; the mobility increases as Rf increases [33]. The Rf was between 0.45 and 0.98 for *nano*MTZ and between 0.50–1.0 for conventional MTZ. Within soil systems, mobility was smaller for NC soil in both formulations (Rf = 0.45 and 0.5, for nanoMTZ and MTZ, respectively). The highest Rf was observed in DS for both formulations (Rf = 0.98–1.0) Among the other soil systems (SC-CT, SC-NT, and SG-MN) Rf ranged from 0.58 to 0.63 for *nano*MTZ and 0.53–0.63 for MTZ (Appendix A). In soil with organic residue addition, Rf values varied between 0.8–0.9 for *nano*MTZ and 0.85–0.98 for MTZ. Within organic materials, Rf values were smaller in black oat straw and corn residue (0.8) for *nano*MTZ, while for conventional MTZ, the lower mobility was obtained in corn and cassava residues (0.85–0.9) (Appendix A).

Mobility in different soil systems and soil with fresh organic residue addition is shown in Figure 5. Thin layer chromatography (TLC) using radiolabeled herbicides is commonly used to verify, qualitatively, the pesticide distribution in soil plates. Both formulations presented lower mobility in soil systems with high SOM content (NC, SC-CT, SC-NT, and SG-MN), compared with DS (Figure 5). In soil with the addition of fresh organic residue, a residue line of 0.5 cm can retain the product and reduce the amount of herbicide distributed in the soil plate. However, it was not enough to reduce the mobility of metribuzin in either formulation (Figure 6).

## 3. Discussion

Sorption is the most important process that defines pesticide behavior in the environment. This process can be described as an interaction between the compound (herbicide) and the sorbent (soil), which controls the bioavailability to plants and environmental fate [34]. The comprehension of this process will aid the proper application of pesticides [35]. Herein, we found low sorption for conventional MTZ and *nano*MTZ. For herbicides, low retention can present two behavior scenarios, (i) greater availability of uptake by plants and other organisms responsible for degradation, and (ii) potential for the other fate process (transport to other environmental compartments/persistence determination). For nanoherbicides, information on the sorption of nanoparticles has been little explored, being an important environmental and plant control indicator to be studied.

The MTZ herbicide is reported to be moderately mobile [36,37,38], as shown by the values described herein (Table 1), and by Mendes et al. [33], Guimarães et al. [39], and Rigi et al. [40]. Here, we showed that the sorption of this herbicide is mainly affected by SOM in different soils, but easily returns to the soil. Metribuzin presented a greater desorption process than sorption rates, becoming available for absorption and transport processes. This was also found by Mielke et al. [41], although availability is dependent on different biochar rates. Mendes et al. [33] indicate deep profile transport for metribuzin (distributed in 30 cm of soil profile), even with the application of organic material on top of the soil. This herbicide was found in the ground and surface water in different countries, presenting environmental risks [42,43,44]. However, when applied in a polymeric nanoformulation, metribuzin safety improved, based on reduced soil mobility in an irrigation field condition and lack of effect in the soil enzymatic activity [14,23].

Here, both formulations presented similar low retentions in different soil systems. MTZ has high solubility (Sw = 10.700 g L^−1^), low lipophilicity (log Kow = 1.75), and is a weak base herbicide (molecular form in pH above pKa = 1), all of which contribute to low sorption in the soil [45,46]. The pH and competition for the ionic sorption sites (CEC, presence of other ions) are not important for the retention of herbicides, such as metribuzin, in the molecular form, as demonstrated by the lack of correlation results (Figure 3). In addition, protonation is not expected to be as important in metribuzin adsorption in the soil [46]. When nanoencapsulated, the formulation has high hydrophilicity due to the surfactant, which provides water affinity to the nanoparticles [47,48], and negative charge (−31 mV), which provides repulsion energy between soil colloids and nanoparticles. On the other hand, high clay content in the soil can offer more sites for the retention of *nano*MTZ in the soil. Despite this, *nano*MTZ showed similar low sorption in the soil compared to conventional MTZ.

Nanoformulations based on PCL can improve the efficacy of the herbicide when applied as a pre-emergent [8,14], and soil availability is important to deliver the herbicide to the plants. Dyianat and Saedian [14] found higher retention for *nano*MTZ than conventional MTZ (40 mm of rain). Nevertheless, Pereira et al. [13] indicated that PCL-atrazine nanoformulation was leached more in soil up to 8 cm in depth (~30%) than conventional atrazine (~20%), with 70 mm of simulated rain. The same formulation based on PCL was tested for MTZ, and conventional MTZ formulation in extreme rain conditions (200 mm in 48 h) presented a high concentration up to 20 cm in depth (~87%), with a great presence in the seed bank in the soil. On the other hand, conventional MTZ is a concern for water contamination, as indicated previously. Therefore, nanoformulation could be a safer alternative than conventional MTZ applications in the field.

Pesticide nanoparticles have not previously been studied in soils of different cropping systems, mainly herbicides applied in the soil, such as MTZ. Here, we observed the increasing sorption potential with the amount of SOM, for both conventional and nanoformulations, in established cropping systems. Humic substances, from the organic matter degradation portion, have high reactivity, a large surface area, and variable composition, interacting with neutral or ionizable molecules [49,50], such as MTZ formulations. Even stable SOM compounds, which increase hydrophobicity in the soil, can sorb triazines and soluble herbicides such as metribuzin [41,51,52]. Therefore, despite the low sorption, soils with consolidated SOM can retain more metribuzin in the soil, in both formulations, reducing the risk of loss. This retention is reversible and can contribute to the absorption process, through the availability of herbicide in the soil.

When the organic matter resource was fresh, the retention in the soil was different, being higher for *nano*MTZ than conventional MTZ. Furthermore, desorption processes occurred more slowly for *nano*MTZ than conventional MTZ in soil systems with added fresh organic matter. With the addition of organic residues in the soil, the interaction between pesticides and soil can be enhanced, mainly by the presence of dissolved organic matter, and consequently change the dynamics of pesticide and nanomaterial dispersion in the environment [53,54]. Dissolved organic matter (DOM) is more present in the soil with fresh materials; and is reduced during the humification process [55]. Nanoparticles can interact with DOM, which can coat nanomaterial surfaces and modify solubility and other properties [56,57,58]. However, it can increase the leaching of pesticides as carriers through soil solution [35]. Another aspect that can increase the effect of nanoformulation sorption is the porous entrapment. Some research has indicated the possibility of pesticide entrapment in biochar and porous soil organic matter [59,60,61]; structures such as nanoparticles in relation to active ingredient alone, can be retained in porous forms in the presence of organic material in the soil. This effect can result in reversible sorption and slow desorption.

In addition to soil retention, short-distance mobility studies can aid understanding the availability of the herbicide to reach the seed bank and the risk of loss in the soil. The Helling and Turner [62] Rf classification considers five classes, with increasing mobility from classes 1 to 5. *Nano*MTZ and conventional MTZ for soil systems were considered slightly mobile according to class 3 (0.35–0.64), similar to other triazine herbicides. On the other hand, both formulations were affected by the presence of fresh organic matter and the Rf values were considered in mobile class 4 (0.65–0.89), according to less organic matter and clay content in the deep soil tested. In deep soil (less SOM) the Rf classification was high mobility (5 class, 0.9–1.0). These results show that fresh residues in poor soil (in relation to the soil systems) are not enough to reduce mobility in the soil. However, all straw types break the herbicide run on soil plates. Other authors also found organic matter as a barrier to triazine herbicides in the soil plates [33,63]. These results can help us understand the effect of organic matter under metribuzin in the soil, independently of the formulation. Even in soils poor in SOM and clay, fresh organic matter played an essential role in metribuzin containment. The combination between SOM retention and a fresh residue barrier to avoid losses can increase the safe application of *nano*MTZ, associated with weed control efficiency for this PCL formulation [16,23].

The effect of fresh organic material, under *nano*MTZ retention, can indicate nanoformulation interference in the retention in soil systems with recent organic matter input. However, the safe application depends on the amount of SOM accumulated after a consolidated system with continuous organic material input. In a system with organic matter addition, this management of the nanoformulation can present slow availability, prolonging the action of the active ingredient in the soil, and reducing losses in the environment. Maintenance of organic matter added in a consolidated soil system can promote the safe application of herbicides in general, mainly nanoherbicides such as *nano*MTZ, due to enhanced retention of molecules in the soil with increased organic matter content. Furthermore, field studies can help us to understand the metribuzin dynamic associated with nanoparticles and consider environmental conditions, as well as the characteristics of soils tested in the laboratory.

## 4. Materials and Methods

### 4.1. Materials

Conventional formulation of metribuzin (Sencor^®^ 480), technical metribuzin (95% purity), and ^14^C-metribuzin (98% purity and 2.3 Bq mg^−1^ of specific activity, American Radiolabeled Chemicals, Inc., St. Louis, MO, USA) and Scintillation solution (Ultima Gold, PerkinElmer, Waltham, MA, USA) were used in the studies. In nanoparticle preparation, caprylic/capric triglyceride (Myritol 318) was purchased from Basf (Basf Co. Ltd., São Paulo, SP, Brazil), poly-ε-caprolactone (PCL) (Mn∼80,000 Da), polysorbate 80 (Mn∼1310 Da), and sorbitane monostearate (Mw = 430.63 g mol^−1^) were purchased from Sigma Aldrich (Sigma-Aldrich, Chem. Co., St. Louis, MO, USA). Soil samples were collected in Paraná and São Paulo State and organic residues were collected in agricultural areas.

### 4.2. Preparation and Characterization of the Nanoformulation

Metribuzin nanoformulation was prepared by the nanoprecipitation method, as published previously [23]. The organic phase consisted of poly-ε-caprolactone (PCL) (100 mg), dissolved in acetone (30 mL) under stirring (55 °C), and mixed under magnetic stirring, after complete PCL dissolution, with Myritol 318 (200 mg), Span 60 (40 mg), and metribuzin (MTZ) (10 mg). The aqueous phase (30 mL) was prepared with the addition of Tween 80 (60 mg), under magnetic stirring. The nanoparticles were formed with a mixture of the organic and aqueous phases, slowly, maintaining the magnetic stirring. The final volume was stirred for 20 min and concentrated in a rotary evaporator until 10 mL (0.4 mg mL^−1^). For the radiolabeled work solution, the ^14^C-metribuzin (95% purity and 2.3 Bq mg^−1^ of specific activity) was added (1.66 × 10^6^ Bq) together with non-radiolabeled MTZ in the organic phase, for the same final concentration (0.4 mg mL^−1^ and 319.6 Bq µL^−1^). The colloidal stability and encapsulation efficiency were indicated previously by Takeshita et al. [23]. Metribuzin nanoparticles were characterized by Takeshita et al. [23] and showed a hydrodynamic size of 289 ± 3 nm (using the DLS technique), zeta-potential of −31.6 ± 0.3, polydispersity index of 0.09 ± 0.02, and spherical morphology by Atomic Force Microscopy (AFM).

For the conventional metribuzin comparison we used the Sencor^®^ 480 (480 g L^−1^), and for the soybean, the recommended dose of 480 g active ingredient (a.i) ha^−1^. For the radiolabeled work solution, the ^14^C-metribuzin was added together with non-radiolabeled MTZ to the solution, in a specific radioactivity amount for each assay, described later in each corresponding section. All *nano*MTZ and conventional MTZ applications used the same dose recommendation.

### 4.3. Soil Collection and Preparation

The soil samples were collected from field areas, in different crop and non-crop systems, in the northern region of Paraná State, Brazil. All samples were collected after the removal of the surface vegetal residues until 0.2 m in depth. The soil samples corresponded to non-crop soil (NC) (22°58′17.7″ S, 50°28′46.1″ W) from an orchard of more than 15 years, cultivated soil with soybean–corn succession in a no-tillage system (SC-NT) (22°58′13.7″ S, 50°28′24.2″ W) for more than 5 years, cultivated soil with soybean–corn succession in a conventional tillage system (SC-CT) (22°58′12.4″ S, 50°27′56.6″ W) for more than 5 years, and sugarcane monoculture soil (SG-MN) (22°58′12.8″ S, 50°28′31.9″ W) for over 15 years. The samples were sieved (2 mm) and stored at 4 ± 2 °C until the studies were carried out. Deep soil (0.8–1 m) (DS) was collected at the center region of São Paulo State, Brazil (22°41′34.7″ S, 47°38′43.0″ W), to test the soil with reduced OM content, used for the added fresh organic materials experiment. For complete information on soil, see Appendix A, and for information on fresh organic materials, see Appendix A. 

### 4.4. Sorption–Desorption Assay in Different Soil Systems

For isotherm determination, the assay was completely randomized, in a 4 × 5 factorial design, with 4 soil (NC, SC-NT, SC-CT, and SG-MN) and 5 MTZ doses (¼ D, ½ D, D, 2 D, and 4 D), with 2 replicates for each formulation (*nano*MTZ and MTZ). All doses were based on the soybean recommendation (480 g a.i. ha^−1^) as the D dose. An additional treatment was used, only for apparent portion calculation, using DS soil, with the objective of elucidating the importance of OM present in the soil profile in the retention of the herbicide. Two replicates were used without soil, only with soil solution, to verify the absence of retention of the formulations in the Teflon tubes.

The sorption–desorption assay was carried out based on the OECD guideline [64]. In a previous test of soil solution + soil centrifugation, we decided to use the proportion 1:2 (adsorbent: solution, *m*/*v*), considering the high clay content in all soils; at an equilibrium time of 24 h, as indicated by Takeshita et al. [23] for MTZ studies. The work solution was formed by radiolabeled *nano*MTZ or conventional and non-radiolabeled MTZ, both formulations added to a CaCl_2_ (0.01 M) solution, considering the amount of active ingredient per soil weight in a hectare, and the radioactivity necessary. 

In a Teflon tube (50 mL), soil content (10 g) was added to the soil solution (19 mL) without herbicide and shaken on the horizontal table (TE 140, Tecnal, Brazil), at 180 rpm for 12 h. Subsequently, 1 mL of work solution with herbicide was added, with shaking at 180 rpm, for 24 h. The final proportion used was 10 g of soil in 20 mL of the soil solution (500 Bq in each tube). For sorption data, all samples were centrifuged at 4500 rpm, for 15 min at 10 °C. Three technical replicates were collected from the supernatant and analyzed using a liquid scintillation counter (Tri-Carb 2910 TR LSA, PerkinElmer, Waltham, MA, USA), for 5 min. The sorption total was considered based on the initial radioactivity applied.

After collecting the sorption aliquots, the supernatant was discarded, and the tubes were reweighed. Again, 20 mL of soil solution was added to the flasks, and the procedure was carried out in the same way as described for sorption evaluation. The total product desorbed was calculated in relation to the amount of product sorbed and the amount of product present in the supernatant.

The sorption coefficient (Kd, mL g^−1^) was calculated using Equation (1):Kd = Cs/Ce(1)
where Cs is the herbicide concentration (mg g^−1^) in the soil after equilibrium and Ce is the herbicide concentration (mg mL^−1^) in the soil solution after equilibrium.

The sorption coefficient normalized by OC (Koc, mL g^−1^) was calculated using Equation (2):Koc = (Kd/OC) × 100(2)
where Kd is the sorption coefficient (mL g^−1^) and OC is the organic carbon present in the soil (%).

The Freundlich sorption coefficient (Kf, mL g^−1^) and 1/n (curve inclination) was calculated using Equation (3):Cs = Kf/Ce^1/n^(3)
where Cs is the herbicide concentration (mg g^−1^) in the soil after equilibrium and Ce is the herbicide concentration (mg mL^−1^) in the soil solution after equilibrium.

### 4.5. Sorption–Desorption Assay as a Function of the Type and Amount of Fresh Organic Material in the Deep Soil

Straw from different cover crops and organic residues was used in this study (forage turnip, black oat, sugarcane, corn waste, and cassava residue). For complete straw information, see Appendix A. The organic materials used were ground, homogenized, and sieved at 2 mm. The fraction used was smaller than 2 mm. These tests were carried out to observe the effect of fresh organic material on metribuzin retention, in a different way to the different crop system soils experiment.

For sorption determination, the assay was completely randomized, with 5 organic residues and unamended soil, with 2 replicates, for each formulation (*nano*MTZ and conventional MTZ). Two more replicates were used without soil, only with soil solution, to verify the absence of retention of the formulations in the Teflon tubes.

All experimental units consisted of 10 g (soil + fresh organic material) and 20 mL of soil solution (1:2, *m/v*), as in the sorption study. The study was conducted in the same way as the retention study in different cropping systems, as described in the previous section.

### 4.6. Soil Thin Layer Chromatography Assay

For mobility information, soil thin layer chromatography was carried out (Soil-TLC), according to the US Environmental Protection Agency OPPTS 835.1210 [65]. The soil samples from different soil systems (NC, SC-NT, SC-CT, and SG-MC) and deep soil + fresh organic material strips in a standard soil were used in the soil-TCL plates (9 cm wide, 15 cm long, and 2 cm thick). The plates were made in duplicate, and ^14^C-metribuzin from MTZ commercial formulation and *nano*MTZ formulation was applied (833.33 Bq) in each plate, to visualize the herbicide mobility. The plates were gently placed in chromatographic chambers containing 100 mL of deionized water until the elution limit line (10 cm above application points) was complete. Subsequently, all plates were dried at room temperature for 48 h. Autoradiograph images were obtained using phosphorescent films, read in radio scanner equipment (Cyclone Plus Phosphor Imager, Model C431200, PerkinElmer Inc., Shelton, CT, USA). Rf (retention factor) values were calculated based on the distance traveled by the herbicide on the plate, using Equation (4):Rf = Dh/Ds(4)
where Dh is the distance from baseline (application point) traveled by herbicide on the plate and Ds is the distance from baseline (application point) traveled by the solvent (water) on the plate (in this case, 10 cm).

### 4.7. Statistical Analysis

The sorption and desorption percentage data of soil systems, and sorption–desorption coefficient, adjusted or not for organic carbon, were submitted to an analysis of variance (ANOVA—two way), as well the sorption–desorption data of *nano*MTZ and conventional MTZ with the addition of different fresh organic materials. When variables were significant, the data were submitted to Tukey’s test or F’s test (*p* < 0.05). Freundlich isotherms were calculated and plotted in Origin 2020 software (Version 9.7.0.185, OriginLab Corporation, Northampton, MA, USA), as were sorption–desorption figures (soil system and straw retention data). Pearson’s correlation was obtained in the RStudio (RStudio 4.1.0 version, Boston USA), as were all statistical tests.

## 5. Conclusions

Our sorption–desorption and mobility experiments suggest low retention and potential mobility of *nano*MTZ and conventional MTZ, as expected for the herbicide metribuzin. In consolidated crop systems, under conventional or no-tillage management, nanoformulation sorption was similar to conventional MTZ, with reversible availability. Sorption and desorption processes were correlated with soil organic matter and organic carbon content. All these soil systems presented similar characteristics, and the SOM was more influential in the retention process. Furthermore, the clay content is important to nanoformulation retention in the soil. Fresh organic residue addition in the deep soil (poorer soil systems) indicated better *nano*MTZ sorption than conventional MTZ. However, TLC quality results showed that fresh residues in poor soil (in relation to the soil systems) are not enough to reduce mobility in the soil, where the organic residues only acted as a physical barrier on the ground. These results suggest that fresh input of organic matter is important to maintain the herbicide in the seed bank zone and reduce leaching risk (slow return to the soil solution), besides contributing to system maintenance. For safe applications of *nano*MTZ in the soil system, continuous organic material addition is essential. Furthermore, we used radiometric techniques through classical protocols, with a focus on cultivation systems, in which nanoMTZ can be inserted early. We reinforce here the importance of testing conventional active ingredients in comparison to new nanoformulations, for better understanding.

## Figures and Tables

**Figure 1 plants-11-03366-f001:**
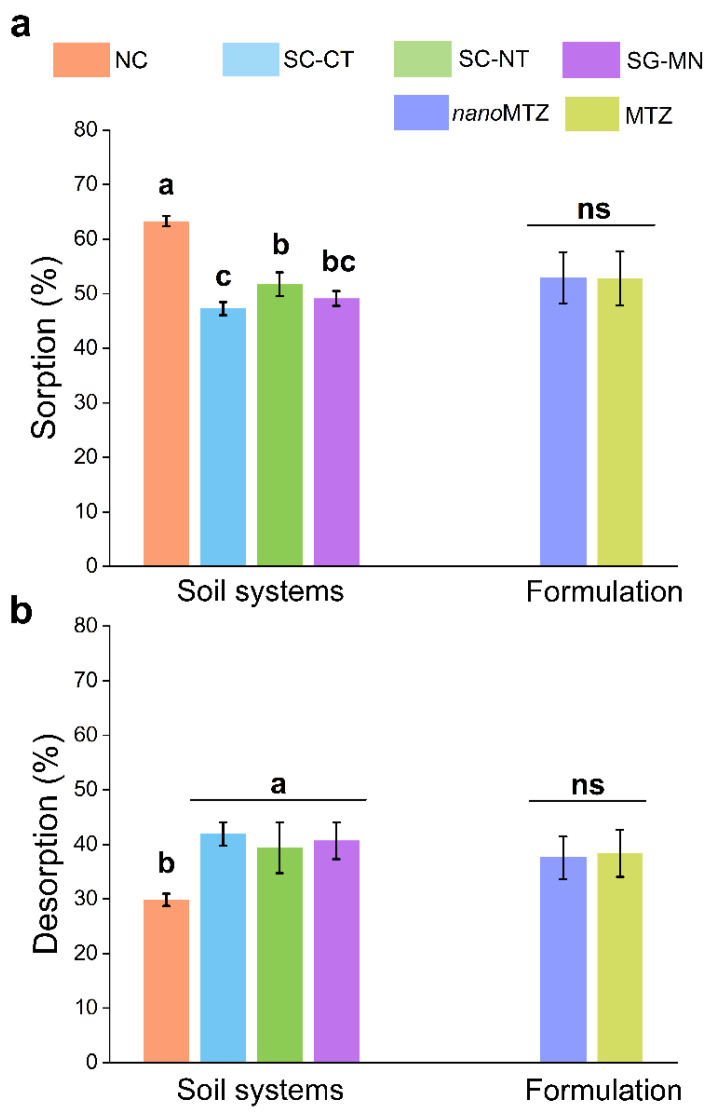
Sorption (**a**) and desorption (**b**) percentages for nanoMTZ and conventional MTZ in different soil systems (non-crop soil—NC, cultivated soil with soybean–corn succession in a no-tillage system—SC-NT, cultivated soil with soybean–corn succession in a conventional tillage system—SC-CT, and sugarcane monoculture soil—SG-MN). Bars represent the standard error of the mean (*n* = 2). Since the interaction between soil system and formulation is not significant, lowercase letters differ between soil systems (regardless of formulation) and ns indicates no differences between formulations by Tukey’s test (*p* < 0.05).

**Figure 2 plants-11-03366-f002:**
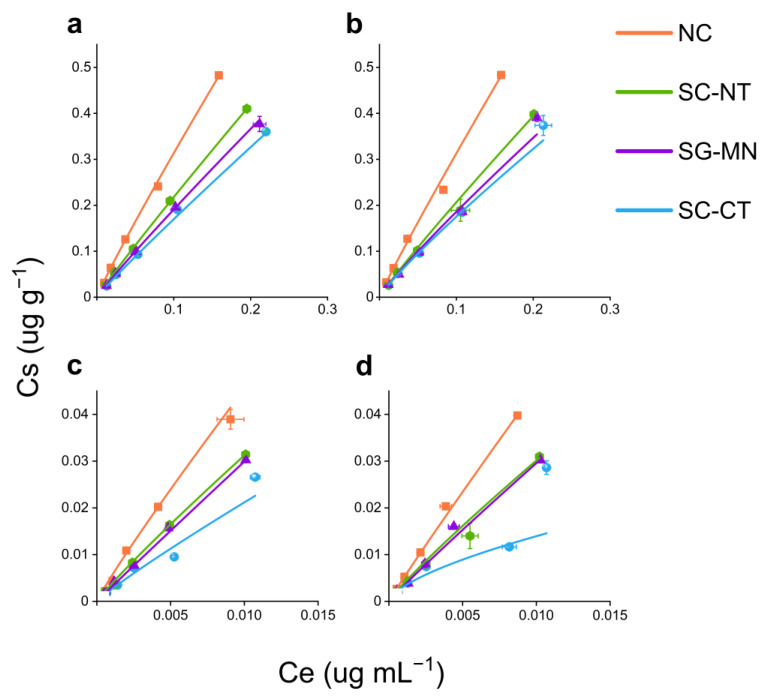
Sorption (**a**,**c**) and desorption (**b**,**d**) isotherms for *nano*MTZ and MTZ, respectively, in different soil systems (non-crop soil—NC, cultivated soil with soybean–corn succession in a no-tillage system—SC-NT, cultivated soil with soybean–corn succession in a conventional tillage system—SC-CT, and sugarcane monoculture soil—SG-MN). Symbols represent Kd values (Cs/Ce) ± mean standard error (*n* = 2). Lines represent the curve for each soil system according to the Freundlich model.

**Figure 3 plants-11-03366-f003:**
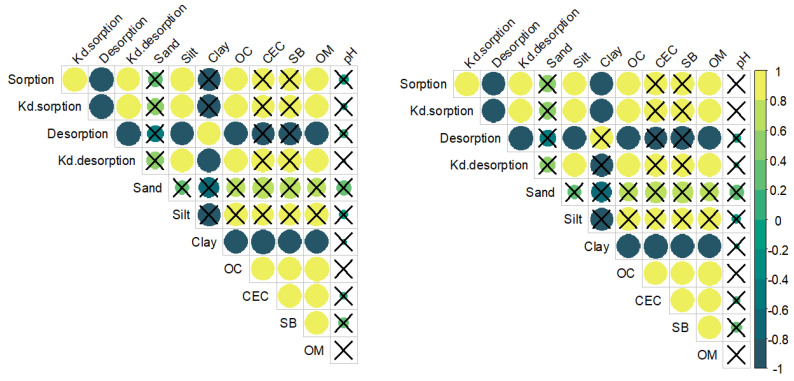
Correlation matrix of sorption–desorption processes and soil system properties for *nano*MTZ (**left**) and conventional MTZ (**right**). Symbols represent Pearson correlation coefficient (*p* < 0.05) and symbols covered by an X represent non-significant correlation coefficients (*p* > 0.05).

**Figure 4 plants-11-03366-f004:**
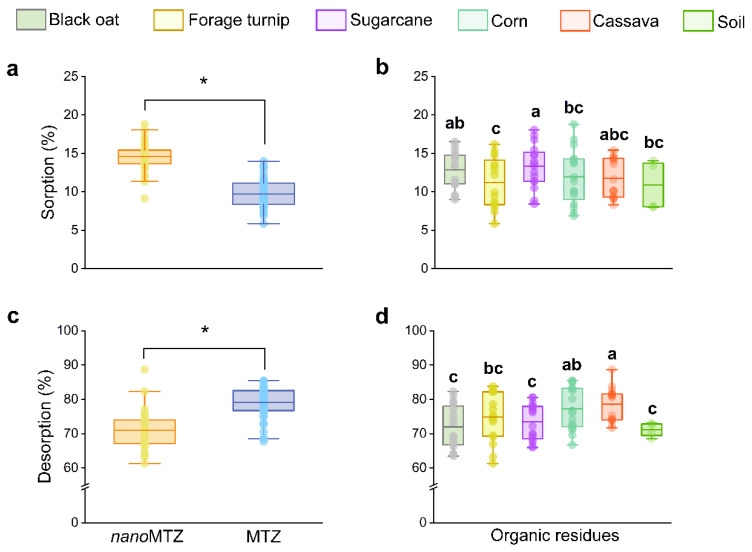
Sorption (**a**,**b**) and desorption (**c**,**d**) of *nano*MTZ and conventional MTZ in soil amended with fresh organic residues. Formulations and residues were compared separately. Asterisks indicate differences between formulations in (**a**,**c**) (F-test, *p* < 0.05), and lowercase letters indicate differences between organic residues in (**b**,**d**) (Tukey’s test, *p* < 0.05), separately. Boxes represents the 25-75% percentiles. The line inside the box indicates de mean, the bars indicates the data range and the circles outside the bars represents outliers.

**Figure 5 plants-11-03366-f005:**
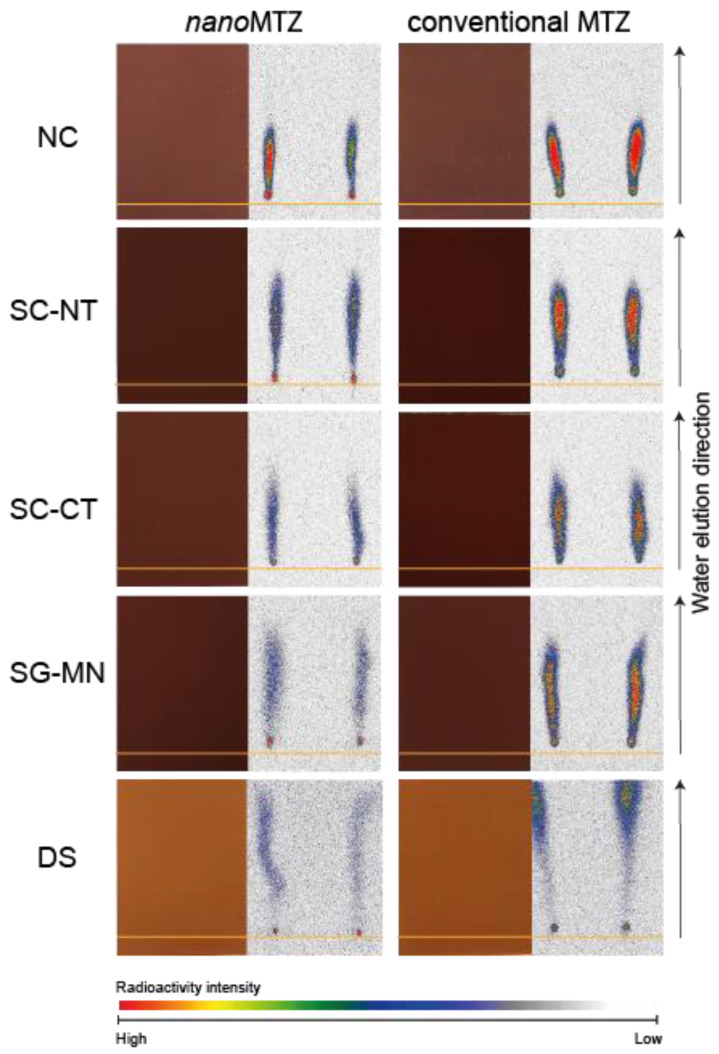
Qualitative mobility data of ^14^C-metribuzin with nanoformulation (left column) and conventional formulation (right column), in different soil systems (non-crop soil—NC, cultivated soil with soybean–corn succession in a no-tillage system—SC-NT, cultivated soil with soybean–corn succession in a conventional tillage system—SC-CT, sugarcane monoculture soil—SG-MN, and deep soil—DS). The orange line is the base herbicide application. Images to the right in each column indicate the soil-TLC plate photographs and images to the left in each column indicate TLC plate autoradiographs after water elution and herbicide mobility in the plates. Colors indicate radioactivity signal intensity.

**Figure 6 plants-11-03366-f006:**
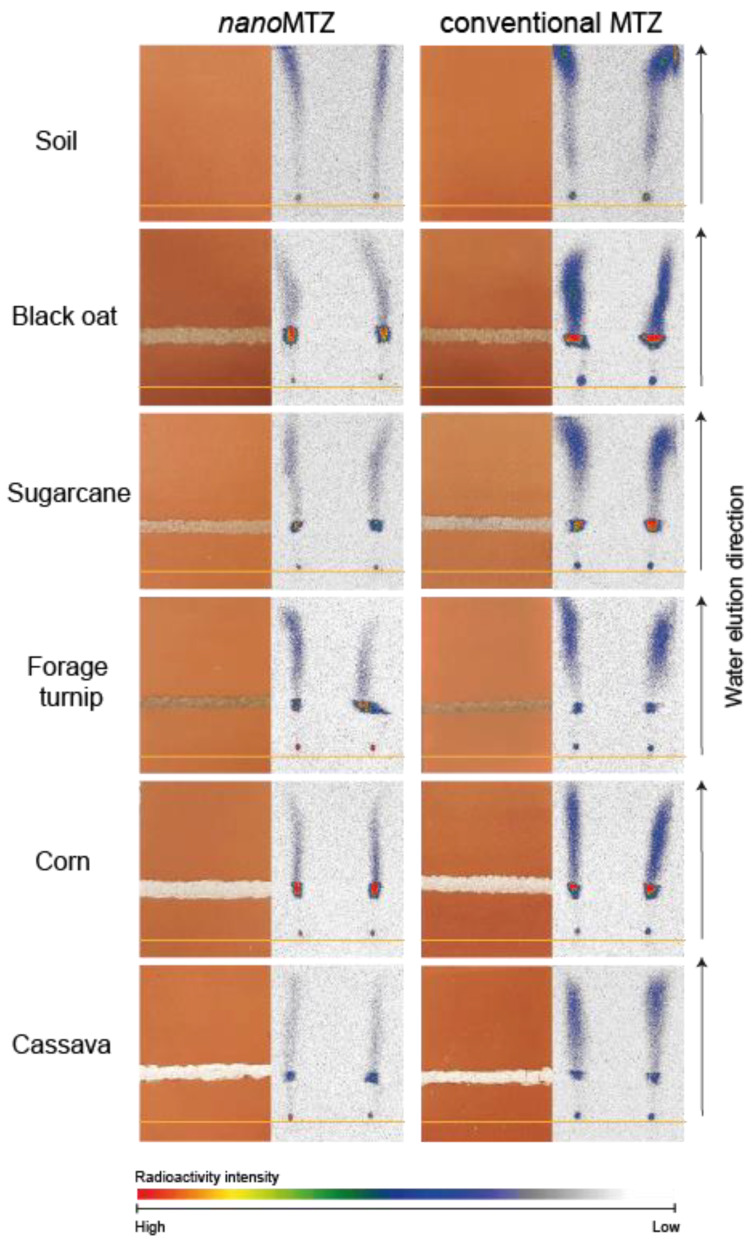
Qualitative mobility data of ^14^C-metribuzin with nanoformulation (**left** column) and conventional formulation (**right** column), in deep soil with fresh organic residue barrier and soil without organic material barrier. The orange line is the base herbicide application. Images to the right in each column indicate the soil-TLC plate photographs and images to the left in each column indicate TLC plate autoradiographs after water elution and herbicide mobility in the plates. Colors indicate radioactivity signal intensity.

**Table 1 plants-11-03366-t001:** ^14^C-metribuzin sorption isotherm parameters for two formulations (*nano*MTZ and conventional MTZ) based on the Freundlich model, in different soil systems (non-crop soil—NC, cultivated soil with soybean–corn succession in a no-tillage system—SC-NT, cultivated soil with soybean–corn succession in a conventional tillage system—SC-CT and, sugarcane monoculture soil—SG-MN. The data indicate the parameter mean ± standard error (*n* = 2). Kd—sorption distribution coefficient. Kf—Freundlich equilibrium constant. 1/n—is the degree of linearity of the isotherm. R² (adj)—adjusted coefficient of determination.

		*nano*MTZ				MTZ			
Parameters		NC	SC-CT	SC-NT	SG-MN	NC	SC-CT	SC-NT	SG-MN
Sorption	Kd (mL g^−1^)	3.4 ± 0.13	1.75 ± 0.01	2.23 ± 0.01	1.96 ± 0.05	3.5 ± 0.08	1.84 ± 0.08	2.07 ± 0.19	1.91 ± 0.11
	Kf (mL g^−1^)	2.66 ± 0.17	1.49 ± 0.03	1.93 ± 0.01	1.66 ± 0.07	2.73 ± 0.08	1.33 ± 0.09	1.76 ± 0.06	1.48 ± 0.16
	1/n	0.929	0.944	0.947	0.940	0.940	0.882	0.931	0.900
	R² (adj)	0.995	0.999	0.999	0.998	0.998	0.996	0.998	0.988
	Hysteresis	-	-	-	-				
Desorption	Kd (mL g^−1^)	5.46 ± 0.51	2.88 ±0.22	3.45 ± 0.19	3.05 ± 0.06	4.91 ± 0.27	2.95 ± 0.26	3.24 ± 0.64	3.34 ± 0.61
	Kf (mL g^−1^)	3.17 ± 1.13	2.79 ± 0.56	1.39 ± 0.54	2.15 ±0.06	3.78 ± 0.22	0.269 ± 0.12	1.89 ± 0.15	2.35 ± 0.51
	1/n	0.922	0.985	0.909	0.919	0.959	0.643	0.899	0.951
	R² (adj)	0.992	0.992	0.945	0.999	0.999	0.922	0.997	0.993
	Hysteresis	0.99	1.04	0.96	0.98	1.02	0.73	1.07	1.06

## Data Availability

Not applicable.

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
