# Peer review of "Availability of Metribuzin-Loaded Polymeric Nanoparticles in Different Soil Systems: An Important Study on the Development of Safe Nanoherbicides"

_plants, 2022, doi:10.3390/plants11233366_

Round 1
Reviewer 1 Report
The authors compared the sorption/desorption/mobility of the pesticide metrebuzin in soil, applied by either using a nano-encapsulated form of the pesticide or the conventional form. The authors do not find significant differences between the mobility of the two formulations in different soil systems. Fresh organic residues content in soil increased the sorption on nano-encapsulated metrebuzin and the authors conclude that continuous addition of organic material to the soil is required for a safe application.
This manuscript is in line with other publications that demonstrate that nano-formulations of pesticides cannot be regarded as safe in an environmental context without consideration of specific conditions in soil.
It is therefore an very interesting manuscript and the overall quality is good, in regard of experimental design as well as writing and language. A minor check for grammar and language errors might be sufficient.
There are only some minor points that should be addressed before the manuscript can be recommended for publication:
The authors compared the mobility of the nano-formulation and conventional form of the pesticide in different soil systems and did not find a significant difference. There are several physiochemical parameters of soil that determine the mobility of colloids, i.e. pH, Na+ concentrations and overall ional strength (parameters shown in .table 5S). Is it possible that pH also influences the mobility of the nano-formulation? A section for the discussion treating the role of physicochemical soil properties should be added.
Figure 1 has some weak points: First, "n=2" is not sufficient to produce a meaningful standard deviation and statistical tests. A minimum of six repeats is required for statistics. Second, it is not clear, what the "a" in fig. 1b stands for. The legend is also not clear when stating: "...lowercase letters differ between soil systems (regardless of formulation)"
Figure 3: "(a)" and "(b)" is referenced in the legend, but not available in the figure. Either add "a" and "b" to the figure or use "left" and "right" in the legend.
Figure 4, b and d are hard to understand. Please describe in the legend what the letters abc stand for and why they are different. Please also indicate in the figure if fig.4 a and b refer to nanoMTZ or MTZ.
First paragraph of results section: Please explain every abbreviation at the first use in text. The explanation for these abbreviations (NC, SG-MN, SC-CT is provided later, but the text cannot be understood by linear reading.
line 79-80: Language: "when applied in soil with the addition of fresh organic matter and an environmentally safe use (?) depends on the system complexity and weed control efficacy."
line 295-297: Language: "On the other hand, both formulations were affected by the presence of fresh organic matter, and were considered in mobile class 4 (0.65-0.89), according to less organic matter and clay content in the deep soil tested." The sentence should be rewritten, as "formulations" should not be "considered 4 (0.56-0.89)" but the Rf value of the formulations.
line 256: "... 8 cm (30%): 8 cm rainfall? This is not a measure of volume. Should rain be provided in mm per surface area? What does 200 mm rain mean, 200mm per hour, per m2 or per hour and m2?
Reviewer 2 Report
Authors have reported the availability of metribuzin loaded polymeric nanoparticles in different soil systems: an important study on the development of safe nanoherbicides following comments should be considered
1. Abstract needs to be rewritten by giving by including backgroud, and introduction.
2. In the introduction authors needd to mention the importance of nanoformulation and nanoparticles in different fields authors may go through different articles they may go through https://doi.org/10.3390/jfb13040207
3. characterizatin technique of the formulation or nanoparticles may be given thats should include SEM, TEM or proper justification needed.
4. HPTLC or HPLC is needed or proper justification needed.
5. Checker for spelling and english grammar.
Reviewer 3 Report
Congratulations to the authors for carrying out such a comprehensive research work. Here are some comments for your consideration
-Lines 28-29: alphabetical sorting of keywords
-Line 87, line 90: Tab. 1S, Fig. 1. Please follow the indications for authors proposed by Plantas. Revise it throughout the text.
-Line 319: A subsection is missing under materials and methods with the materials and reagents used.
-Line 320: explain in detail how the characterization of the nanoparticles was performed, as well as provide a microscopic image of their formation.
-Line 343: provide the coordinates of the plots where the samples were collected
-Line 335: specify the meaning of active ingredient (a.i.) previously.
-Line 412: (m/v) in italics
Round 2
Reviewer 2 Report
Comments incorporated